# Retinal Ganglion Cell Survival and Axon Regeneration after Optic Nerve Transection is Driven by Cellular Intravitreal Sciatic Nerve Grafts

**DOI:** 10.3390/cells9061335

**Published:** 2020-05-27

**Authors:** Zubair Ahmed, Ellen L. Suggate, Ann Logan, Martin Berry

**Affiliations:** Neuroscience and Ophthalmology, Institute of Inflammation and Ageing, University of Birmingham, Birmingham B15 2TT, UK; ellie.jenner@gmail.com (E.L.S.); a.logan@bham.ac.uk (A.L.); m.berry@bham.ac.uk (M.B.)

**Keywords:** axon regeneration, CNS, retinal ganglion cells, optic nerve, optic nerve transection, neuroprotection, neurotrophic factors, peripheral nerve grafts

## Abstract

Neurotrophic factors (NTF) secreted by Schwann cells in a sciatic nerve (SN) graft promote retinal ganglion cell (RGC) axon regeneration after either transplantation into the vitreous body of the eye or anastomosis to the distal stump of a transected optic nerve. In this study, we investigated the neuroprotective and growth stimulatory properties of SN grafts in which Schwann cells had been killed (acellular SN grafts, ASN) or remained intact (cellular SN grafts, CSN). We report that both intravitreal (*ivit*) implanted and optic nerve anastomosed CSN promote RGC survival and when simultaneously placed in both sites, they exert additive RGC neuroprotection. CSN and ASN were rich in myelin-associated glycoprotein (MAG) and axon growth-inhibitory ligand common to both the central nervous system (CNS) and peripheral nervous system (PNS) myelin. The penetration of the few RGC axons regenerating into an ASN at an optic nerve transection (ONT) site is limited into the proximal perilesion area, but is increased >2-fold after *ivit* CSN implantation and increased 5-fold into a CSN optic nerve graft after *ivit* CSN implantation, potentiated by growth disinhibition through the regulated intramembranous proteolysis (RIP) of p75^NTR^ (the signalling trans-membrane moiety of the nogo-66 trimeric receptor that binds MAG and associated suppression of RhoGTP). Mϋller cells/astrocytes become reactive after all treatments and maximally after simultaneous *ivit* and optic nerve CSN/ASN grafting. We conclude that simultaneous *ivit* CSN plus optic nerve CSN support promotes significant RGC survival and axon regeneration into CSN optic nerve grafts, despite being rich in axon growth inhibitory molecules. RGC axon regeneration is probably facilitated through RIP of p75^NTR^, which blinds axons to myelin-derived axon growth-inhibitory ligands present in optic nerve grafts.

## 1. Introduction

Axons readily regrow in the damaged peripheral nervous system (PNS) but fail to do so in the injured central nervous system (CNS). This differential injury response is attributed to the presence in the PNS of neurotrophic factors (NTFs) derived from Schwann cells [1,2,3] and the growth compatible substrate of their laminin-rich basal lamina tubes and, in the injured CNS, to a lack of NTF support and the presence of axon growth-inhibitory ligands. The latter comprise molecules released from meningeal fibroblasts migrating into the incipient scar, reactive astrocytes, and degraded CNS myelin. Paradoxically, the potent axon growth-inhibitory ligands, chondroitin sulphate proteoglycan (CSPG), and myelin-associated glycoprotein (MAG), are components of PNS extracellular matrix (ECM) and myelin, respectively [4,5,6] which, after binding to GT1b [7] and/or the Nogo-66 receptor (NgR1) [8], signal axon growth collapse through p75^NTR^ transmembrane NgR1 co-receptor activation of the RhoGTP/LIMK/cofilin pathway [9].

The failure of PNS CSPG/MAG to inhibit regeneration may be explained by the efficient removal of ECM/myelin debris by macrophages and Schwann cells [10,11,12], supported by the observation that axon growth inhibition in the PNS has been overcome by degrading CSPG with chondroitinase ABC treatment [13]. The possibility that Schwann cell basal lamina tube laminin may override the growth inhibitory activity of MAG and both Schwann cell-derived matrix metalloprotease (MMP)-2 degradation of PNS inhibitors [14] and NTF-induced activation of the PI3K/Akt pathway [15,16,17,18] may all lead to inactivation of down-stream glycogen synthase kinase (GSK3)-β and the triggering of regulated intramembranous proteolysis (RIP) of the p75^NTR^/NgR1 signalling co-receptor [19,20], thereby protecting against axon growth cone collapse and preserving their mobility.

Although PNS axons readily regenerate into cellular sciatic nerve (SN) (CSN) grafts, they grow sparingly and for very short distances into acellular SN (ASN) grafts [21,22,23] and in-growth failure persists after blocking Schwann proliferation in the proximal PNS stump [21]. Nonetheless, PNS axons do enter ASN grafts after the removal of CSPG/MAG-rich debris by Schwann cells and macrophages; however, pre-degeneration before freeze–thawing of the SN promotes some axon regeneration into ASN [14,24], suggesting that inhibitory ligands are preserved in ASN. Moreover, repopulation of ASN by Schwann cells after migration from the proximal peripheral nerve (PN) stump [21,25,26], or pre-seeding in culture [27] or sub-perineural injection [28] promotes entry of PNS axons; observations that support the concept that Schwann cells are leaders not followers of PNS axons regenerating into Schwann cell basal lamina tubes [25,29,30].

Retinal ganglion cell (RGC) axons regenerate into CSN but not ASN grafts anastomosed to the proximal optic nerve stump after optic nerve transection (ONT) [31,32,33]. Similarly, RGC axons regenerate across an optic nerve crush site after intravitreal (*ivit*) CSN but not *ivit* ASN implantation [34,35,36]. Interestingly, RGC neuroprotective factors are released from both CSN and ASN at ONT sites and promote RGC survival after retrograde transport to RGC [32]. RGC axons are probably attracted into the basal lamina tubes of CSN by NTF secreted by resident Schwann cells [1] and readily elongate over their plasmalemma and the laminin rich inner basal lamina tube surface [37,38,39]. Whereas failure of RGC axons to enter ASN grafts may be explained by an absence of Schwann cell-derived NTF and the persistence of CSPG/MAG inhibitory ligands, i.e., the constitution of ASN is essentially similar to that of the optic nerve through which axons also will not grow after injury. In this study, we tested this hypothesis by evaluating the growth of RGC axons into ASN grafted onto a proximal optic nerve stump after *ivit* CSN implantation, predicting that CSN-derived NTF will induce disinhibited growth of RGC axons into the inhibitory environment of an ASN graft, as they do through an optic nerve crush site [19,20,32]. We also investigated the RGC neuroprotective properties of ASN by comparing their neurotrophic potency as *ivit* as well as optic nerve grafts and evaluate the contribution of reactive Mϋller cells/astrocytes and macrophages to these responses.

## 2. Materials and Methods

### 2.1. Animals

We used adult male Fischer rats (Charles River, Maidstone, UK) weighing 170–250 g for all experiments in this study. Animals were fed a commercial diet and water ad libitum under controlled conditions (22 ± 2 °C, 55% ± 5% humidity, and a 12-h light/12-h dark cycle). All surgical procedures were licensed by the UK Home Office and approved by the University of Birmingham’s Animal Welfare and Ethical Review Board (PPL: 70/08542; date of approval: 12/03/2015). All animal surgeries were carried out in strict accordance with the guidelines of the UK Animals Scientific Procedures Act, 1986, the Revised European Directive 1010/63/EU, and conformed to the guidelines and recommendations of the use of animals by the Federation of the European Laboratory Animal Science Associations (FELASA). Every effort was made to reduce the number of animals employed and to minimize animal discomfort. Pre- and post-operative analgesia was used as standard and with guidance from the named veterinary surgeon.

### 2.2. Experimental Design

All animals were randomly assigned to experimental groups with the experimenter masked to the treatment conditions. The optic nerve of adult male Fischer rats was crushed bilaterally [20,34,35,40,41,42,43] and CSN and ASN implanted *ivit* and/or anastomosed to the cut end of the transected optic nerve to study their effects on RGC survival and axon regeneration. Unless otherwise stated, experimental groups comprised 8 rats in each group (i.e., 16 optic nerves and 16 retinae/group): (i), after Sterispon (S) plugging of a scleral incision through the retina into the vitreous body—sham *ivit* implantation group (Control (CON)/*ivit*_S_/ONT); (ii) an ASN was implanted *ivit* immediately after ONT—*ivit*_ASN_/ONT group; (iii) after sham *ivit* implantation, an ASN was anastomosed to the proximal ONT site—*ivit*_S_/ONT_ASN_ group; (iv), an ASN was implanted *ivit* immediately after ONT and an ASN was anastomosed to the proximal ONT site—*ivit*_ASN_/ONT_ASN_ group; (v), a CSN was grafted *ivit* immediately after ONT—*ivit*_CSN_/ONT group; (vi), a CSN was implanted *ivit* after ONT and an ASN graft anastomosed to the proximal ONT stump—*ivit*_CSN_/ONT_ASN_ group; (vii), an ASN was implanted *ivit* and a CSN graft immediately anastomosed to the proximal ONT stump—*ivit*_ASN_/ONT_ASN_ group; (viii), a CSN was implanted *ivit* and also immediately anastomosed to the proximal ONT stump—*ivit*_CSN_/ONT_CSN_ group. Uninjured rats were used to control for baseline parameters where indicated—intact groups (Figure 1). Optic nerves and retinae were processed for immunohistochemistry and western blot at 21 days after surgery, as described below.

### 2.3. Preparation of Sciatic Nerve (SN) Grafts

All grafts were prepared from adult Fischer rat SN segments from which the lumbar dorsal and ventral roots rostrally and the distal nerve branches from the origin of the peroneal nerve caudally were removed. SN segments for anastomosis to the proximal ONT stump were prepared from 5 mm lengths which either remained intact as CSN or were freeze/thawed x3 to kill all Schwann cells as ASN, as described by us [44]. For *ivit* implantation, 2 mm lengths of CSN and ASN were prepared as pellets by teasing in phosphate-buffered saline (PBS).

After immunohistochemistry with the antibody marker p75^NTR^ for Schwann cells and laminin for Schwann cell basal lamina tubes (Table 1), we confirmed that p75^NTR+^ Schwann cells, with typical spindle morphology, were abundant in CSN (Appendix A) but absent in ASN (Appendix A) at 21 days after surgery, and that Schwann cell laminin^+^ basal lamina tubes were preserved in both CSN and ASN (Appendix A, respectively).

### 2.4. Surgery

Details of the grafting and optic nerve surgery are given by us previously [32,33,34]. Briefly, *ivit* implantation of CSN and ASN pellets was performed by an intra-orbital approach through the upper lid. A vertical 1 mm incision in the superior hemi-scleral surface allowed insertion through the retina into the vitreous body of the grafts which were retained by absorbable gelatin sponge (Sterispon, Allen and Hanbury, London, UK) plugs. For anastomosis of 5 mm long CSN and ASN to ONT, the optic nerve was transected 2 mm from the lamina cribrosa and grafts sutured to the proximal cut end of the optic nerve with 10/0 resorbable sutures (Ethicon, Somerville, NJ, USA). For intra-orbital ONT, the optic nerve was exposed through the upper lid, transected, within the dural sheath 2 mm from the lamina cribrosa and the cut ends sutured with 10/0 resorbable sutures (Ethicon).

### 2.5. Fluorogold (FG) Labelling, Retinal Wholemounts, and Retinal Ganglion Cell (RGC) Counting

RGC were retrogradely labelled with FG (Cambridge Bioscience, Cambridge, UK) by injecting 2 μL of a 4% FG solution into the optic nerve, proximal to the ONT site, 48 h before enucleation [34,35,42,43]. Retinae (*n* = 16/group) were immersion-fixed for 2 h in 4% formaldehyde (TAAB Laboratories, Berkshire, UK), adhered onto Superfrost Plus microscope slides (VWR international, Leicestershire, UK) as whole mounts and cover slipped in Vectashield fluorescent mounting medium (Vector Laboratories, Peterborough, UK). Retinae were randomized and examined using an epifluorescent microscope (Zeiss, Hertfordshire, UK) and photographed with an AxioCam HRc digital camera controlled by Axiovision 4 software (all from Zeiss). FG^+^ RGC were counted using the automated function in ImagePro Software, version 6.0 (Media Cybernetics, Bethesda, USA) from images of 12 rectangular areas (0.36 × 0.24 mm), 3 from each quadrant of each retina, placed at standard radial distances from the centre of the optic disc at the inner (1/6 eccentricity), mid-periphery (1/2 eccentricity), and peripheral retina (5/6 eccentricity) and RGC densities computed as mean RGC density/mm^2^ (*n* = 8 rats/group, 16 retinae/group) [41].

### 2.6. Tissue Harvesting, Sectioning, and Immunohistochemistry

Animals were killed by CO_2_ overdose followed by perfusion fixation with 4% formaldehyde (TAAB Laboratories) [42,43]. Eyes and optic nerves were removed and immersed in 4% formaldehyde before cryoprotection in graded concentrations of sucrose (10%, 20%, and 30% in PBS). Tissues were embedded in OCT embedding medium (R.A. Lamb Laboratory Supplies, Eastbourne, UK) and stored at −80 °C until required. Parasagittal sections of eyes and longitudinal sections of the optic nerve (both 15 μm thick) were cut on a cryostat (Bright Instrument Company, Huntingdon, UK) and adhered onto Superfrost^®^ Plus electrostatically charged microscope slides (VWR International), air-dried and stored at −20 °C until required.

After thawing at room temperature for 30 min, at least 6 sections/rat/antibody, from equivalent areas of the eye globe (*n* = 8 rats/group (i.e., 16 retinae/group)) were washed 2× in PBS, permeabilized in PBS containing 0.1% Triton X-100 for 10 min, washed 3× in PBS and blocked in 3% bovine serum albumin (BSA, Sigma) in PBS containing 0.05% Tween-20 (PBS-T-BSA) for 1 h at room temperature. Sections were incubated overnight (16–18 h) at 4 °C with the relevant primary antibody (Table 1) diluted in PBS-T-BSA. For fluorescent immunohistochemistry, sections were washed 3× in PBS and incubated with relevant fluorescent Alexa 488 or Texas Red-labelled fluorescent secondary antibodies (Table 1) before washing 3× in PBS and mounting in Vectashield fluorescent mounting medium (Vector Laboratories). Controls including sections incubated with primary antibody omitted were included in each run and were used to set the background threshold before image capture.

For immunodetection using the 3,3′-diaminobenzidine (DAB) substrate, sections were incubated overnight in primary antibody, washed, and incubated with HRP-labelled secondary antibody (Vector Laboratories) for 1 h at room temperature. After washing in PBS 3×, sections were incubated in Avidin-Biotin complex (Vector Laboratories) for 30 min, followed by 3× washes in PBS. Immuno-positive staining was developed for 3 min using the DAB substrate (Vector Laboratories). Sections were then washed in PBS, followed by tap water, dehydrated through a graded series of alcohols, cleared in Histoclear, and mounted in Vectamount medium (Vector Laboratories) and images captured using AxioCam HRc and Axiovision 4 software attached to either a fluorescent or light microscope (all from Zeiss).

### 2.7. Evaluation of Müller Cell/Astrocyte and Macrophage Responses

Müller cell/astrocyte activation was monitored using immunohistochemistry for glial fibrillary acidic protein (GFAP) as described above (Table 1). Müller cell activation was quantified in GFAP stained retinal sections as described previously [42]. Briefly, a 250 µm long sampling line set orthogonal to the radial plane through the middle of the inner plexiform layer was used to count the number of GFAP^+^ Müller cell processes and mean counts were then calculated for each condition (*n* = 3 retinal sections/rat/condition (*n* = 8 rats/group (i.e., 24 retinal sections/group)) and expressed as mean ± SEM.

Macrophage responses to the various grafting paradigms were assessed in retinal and ONT grafting sites using ED1 (Table 1) as a marker.

### 2.8. Detection of Regenerating Retinal Ganglion Cell (RGC) Axons

#### 2.8.1. Anterograde Tracing

Rhodamine B (RhB, Sigma, Poole, UK) (5 μL of a 2% solution diluted in 2% dimethyl sulfoxide (DMSO)) was injected into the vitreous body 48 h before optic nerve harvesting at 21 days. Animals (*n* = 6 rats/group (i.e., 12 optic nerves/group)) were killed by CO_2_ overdose, perfusion fixed in 4% formaldehyde (TAAB Laboratories) and their optic nerve cryoprotected in a graded series of sucrose, embedded in OCT (R.A. Lamb) and 15 μm thick longitudinal sections cut on a cryostat, as described above. Optic nerve sections were then washed 2× in PBS, mounted using Vectashield fluorescent mounting medium (Vector Laboratories) and RhB^+^ axons visualized using an epifluorescent microscope (Zeiss).

#### 2.8.2. Growth-Associated Protein 43 (GAP43) Immunohistochemistry

Longitudinal optic nerve sections were washed 2× in PBS, blocked in PBS-T-BSA for 1 h, and incubated with primary GAP43 antibody overnight (Table 1). Sections were then washed 3× in PBS and incubated with Alexa 488 donkey anti-mouse IgG (Invitrogen) for 1 h at room temperature before washing 3× in PBS and mounting in Vectashield fluorescent mounting medium (Vector Laboratories). Scarring at the ONT site was monitored by exposing GAP43-stained sections to GFAP and laminin antibodies and processing as above.

### 2.9. Quantification of Axon Regeneration

Regenerating GAP43^+^ or RhB traced axons were counted in optic nerve sections using previously published methods [45,46]. GAP43^+^ axons were counted at 400, 800, and 1200 μm distal to the ONT sites in 4 longitudinal optic nerve sections (*n* = 8 rats/group (i.e., 16 optic nerves/group)). The diameter (d) of each optic nerve at the 3 specific counting sites (d1, 2, 3) was also measured using Axiovision Software (Zeiss) and the number of axons/mm of optic nerve width calculated. The average total number of axons (Σ_ad1,2,3_) in the optic nerve of radius r, extending a distance *d* calculated by summing over all sections of thickness *t* (15 μm):Σad1, 2, 3 = πr2 × (average number of axons/mm width)/(section thickness (0.015 mm))(1)

### 2.10. Separation of Retina and Vitreous Body for RNA Isolation and Oncomodulin PCR

The vitreous body was isolated before RNA extraction and qPCR for oncomodulin as described previously [47]. Briefly, retinae together with their vitreous bodies were isolated, quickly flat-mounted on nitrocellulose filters, transferred to Whatman filter paper. The vitreous body sticking on top of the retina was carefully separated from the retina and transferred to an Eppendorf tube before RNA extraction using the RNeasy-kit (Qiagen, Watford, UK) according to the manufacturer’s protocol. Oncomodulin expression was quantified using the Synergy Brands (SYBR) green reverse transcription (RT) PCR using the LightCycler instrument. Briefly, 100 ng of total RNA from each vitreous was reverse transcribed and the cDNA amplified with specific primers for 50 cycles according to the manufacturer’s protocol. All reactions were performed in duplicate using *n* = 4 independent samples/group/condition (from *n* = 4 different eyes). Quantitative analysis was performed using the LightCycler software and the specificity of the PCR products in each analysis was verified with the melting-curve analysis feature of LightCycler software. Two different sets of primers were used for OM gene amplification: custom made primers, Rn_Ocm_1_SG QuantiTect^®^ and GAPDH_RN_Gapd_1 SG QuantiTect^®^ (Qiagen); and a second set, 5′-AAGACCCAGACACCTTTGAAC-3′ and 5′-GAACTTCTGTAGGAAATACTTGAGC-3′. Rat Morris-hepatoma cells (ATCC CRL-1601), which we confirmed to express high levels of oncomodulin, were used as a positive control.

### 2.11. Ciliary Neurotrophic Factor (CNTF) mRNA Expression and ELISA in Retinae

Total RNA from retinae (*n* = 4 retinae/group) was extracted as described above and subjected to qRT-PCR using a pre-validated rat CNTF primers (ThermoFisher Scientific, cat no. 4331182, Rn00755092_m1). Rat GAPDH primers were used as controls (ThermoFisher Scientific, cat no. 4331182, Rn01775763_g1).

After extraction of total protein by homogenization in ice-cold lysis buffer (20 mM Tris-HCl, pH7.4, 150 mM NaCl, 1 mM EDTA, 0.5 mM EGTA, and 1% NP-40, all from Sigma), a CNTF ELISA kit was used to determine the concentration of CNTF in retinal lysates, according to the manufacturer’s instructions (R&D Systems Europe, Abingdon, UK).

### 2.12. Protein Extraction and Western Blotting

Animals were killed by CO_2_ overdose, retinae/optic nerves immediately harvested, and total protein extracted by homogenization in ice-cold lysis buffer (20 mM Tris-HCl, pH 7.4, 150 mM NaCl, 1 mM EDTA, 0.5 mM EGTA, and 1% NP-40, all from Sigma). Samples were normalized for protein content using a colorimetric DC protein assay (Bio-Rad, Hemel Hempstead, UK), resolved on 12% SDS-PAGE gels and processed for western blotting as previously described [41,42,43]. Western blots were probed overnight at 4 °C with relevant antibodies (Table 1) and bands stained with an appropriate HRP-labelled secondary antibody (Table 1) (GE Healthcare, Buckinghamshire, UK) and detected using an enhanced chemiluminescence system (ECL) (GE Healthcare). Western blots were stripped and probed with a rabbit monoclonal antibody to GAPDH (Cell Signaling Technology, Danvers, MA, USA) at 1:1000 dilution and used as a loading control.

### 2.13. Evaluation of Neurite Outgrowth Inhibior-A (Nogo-A), Nogo-C, Myelin Associated Glyocprotein (MAG), and Chondroitin Sulphate Proteoglycan (CSPG) Content of Optic Nerve, CSN, and ASN

Nogo-A, Nogo-C, MAG, and CSPG content of optic nerves, CSN, and ASN were analysed by Western blot using tissues lysates exposed to specific antibodies (Table 1). Blots (*n* = 3/sample, 3 independent repeats) were then quantified by densitometry as described later.

### 2.14. Rho Activation Assay

GTP bound Rho was detected using a Rho activation assay kit according to the manufacturer’s instructions (Upstate Biotechnology, Milton Keynes, UK) and as described by us previously (e.g., [20,41,42,48].

### 2.15. Densitometry

Western blots were quantified by densitometry after scanning blots into Adobe Photoshop as TIFF files and analysed using built-in gel plotting macros in ImageJ (NIH, Maryland, USA), as described by us previously (e.g., [20,41,42,43]) and the integrated densities of relevant bands calculated from 3 separate blots from 3 independent experiments.

### 2.16. Statistical Analysis

Significant differences between sample means were calculated using GraphPad Prism (GraphPad Software Inc., Version 4.0, CA, San Diego, USA) by one-way analysis of variance (ANOVA) followed by post-hoc testing with Dunnett’s method. A *p*-value of <0.05 was considered significant.

## 3. Results

### 3.1. Retinal Ganglion Cell (RGC) Survival

Details of the RGC counts at 21 d after grafting are summarized in Table 2 and Figure 2A,B. (i), After ivit grafting: RGC numbers declined significantly by ~90% in the CON/ivit_S_/ONT group to 11% of the value of the Intact controls and did not recover in the ivit_ASN_/ONT (9%) implying that ivit_ASN_ grafts are not neuroprotective. By contrast, ivit_CSN_/ONT implants were ~25% RGC neuroprotective. (ii), After optic nerve grafting: Only 13% RGC survived in the ivit_S_/ONT_ASN_ group, not significantly different from CON/ivit_S_/ONT group values, although significant RGC neuroprotection of 15% was recorded in the ivit_S_/ONT_CSN_ group. (iii) After both ivit and optic nerve grafting: The greatest RGC neuroprotection of ~58% was achieved in the ivit_CSN_/ONT_CSN_ group, i.e., >50% RGC survival compared to the numbers of RGC surviving after ivit_CSN_/ONT (25%) and ivit_S_/ONT_CSN_ (15%), suggesting that combined ivit_CSN_/ONT_CSN_ grafting had a synergistic effect on RGC survival. Whereas, in the ivit_CSN_/ONT_ASN_ group, RGC survival was similar to that seen in the ivit_CSN_/ONT (28% and 25%, respectively), indicating that combined ivit_CSN_/ONT_ASN_ grafting had no additive effect above that offered by single ivit_CSN_ grafting. These results suggest that ivit_CSN_ grafts promote significant RGC survival after ONT whilst a combination of ivit_CSN_/ONT_CSN_ is additive and promotes the survival of 58% of RGC.

### 3.2. The Effect of Cellular (CSN) and Acellular Sciatic Nerve (ASN) Grafts on Müller Cell/astrocyte Activation at 21 Days After Grafting

(i), Controls: In the CON/intact group, GFAP^+^ staining was restricted to astrocyte and Müller cell end-feet in the fibre layer and ganglion cell layer (GCL); no GFAP^+^ Müller cell processes were seen in the retina (Figure 3A,J). In the CON/ivit_S_/ONT group there was muted but significant (*p* < 0.001) generalized retinal glia reactivity, which became more intense at the scarred sham implantation site where no Sterispon remained compared to CON/intact groups (Figure 3B,J). (ii), optic nerve grafts: The level of GFAP^+^ Müller cell/astrocyte activation in the ivit_S_/ONT_ASN_ (Figure 3C,J) and ivit_S_/ONT_CSN_ groups (Figure 3D,J) was similar to that in the CON/ivit_S_/ONT group. (iii), Ivit grafts: In the ivit_ASN_/ONT group (Figure 3E,J), Müller cell/astrocyte activation was similar to that observed for CON/ivit_S_/ONT and other groups of animals receiving optic nerve grafts; however, in the ivit_CSN_/ONT (Figure 3F,J) and ivit_CSN_/ONT_ASN_ (Figure 3G,J) groups there was a significant increase (1.8-fold; *p* < 0.0001) in Müller cell/astrocyte activation compared to CON/ivit_S_/ONT or other groups of animals receiving ON grafts. Müller cell/astrocyte activation in the ivit_ASN_/ONT_CSN_ group (Figure 3H,J) was not significantly different to CON/ivit_S_/ONT or other groups of animals receiving ON grafts but in ivit_CSN_/ONT_CSN_ (Figure 3I,J) groups there was a significant (*p* < 0.0001) >1.6-fold increase in Müller cell/astrocyte activation over the entire retina compared ivit_CSN_/ONT and ivit_CSN_/ONT_ASN_ groups.

The switch to axon regeneration in RGC to lens injury was closely correlated with Müller cell/astrocyte activation and injury-induced stimulation and release of CNTF from retinal glia were shown to switch mature RGCs to a regenerate state [47]. In our injury paradigms, there was a slight increase in CNTF mRNA in the retina in CON/ivit_S_/ONT and in ivit_ASN_ paradigms but a significant increase in all ivit_CSN_ paradigms over the first seven days (Appendix A).

These results indicate that ivit_CSN_ grafts induce greater levels of Müller cell/astrocyte activation throughout the retina compared to optic nerve grafts, with the greatest levels of Müller cells/astrocyte activation observed with simultaneous ivit_CSN_/ONT_CSN_ grafts. The levels of CNTF mRNA also correlates with these increases in Müller cell/astrocyte activation in the ivit_CSN_ group.

### 3.3. Macrophage Activation after Grafting

(i), Controls: No ED1^+^ macrophages were present in the retinae/vitreous bodies of CON/intact (Figure 4A). In the CON/ivit_S_/ONT group a few macrophages were seen in the retina at the sham implantation site where the scleral incision had healed and the Sterispon plug had disappeared, but there was no generalised invasion of macrophages into the vitreous body or retina (Figure 4B). (ii), After optic nerve grafting: In the ivit_ASN_/ONT and ivit_S_/ONT_ASN_ groups, no ED1^+^ macrophages were found in the retina/vitreous body (Figure 4C,D) (iii), After ivit grafting: In ivit_CSN_/ONT, ivit_CSN_/ONT, ivit_CSN_/ONT_ASN,_ and ivit_CSN_/ONT_CSN_ groups, ED1^+^ macrophages had invaded the retina/vitreous body but were absent in ivit_ASN_/ONT_CSN_ groups (Figure 4E–I). ED1^+^ macrophages had also accumulated in both ivit_CSN_ (Figure 4J) and ivit_ASN_ implants (Figure 4K). At the site of anastomosis in the optic nerve, similar levels of ED1_+_ immunoreactivity (red) was observed in ivit_CSN_/ONT_CSN_, ivit_CSN_/ONT_ASN,_ and ivit_CSN_/ONT_CSN_ groups (Figure 4L–N).

Despite the detection of large numbers of ED1^+^ macrophages in ivit_CSN_ paradigms, there was no difference in oncomodulin mRNA, in any of the ivit_CSN_ groups compared to CON/Intact groups (Appendix A). Oncomodulin is a macrophage-derived factor that has been shown to promote significant RGC survival and axon regeneration after optic nerve injury [49,50,51,52].

These results suggest that only ivit_CSN_ grafts promoted ingress of macrophages into the eye despite macrophages being present in both CSN and ASN implants and that OM levels do not change in ivit_CSN_ paradigms compared to CON/Intact groups.

### 3.4. Presence of Axon Growth Inhibitory Ligands in Cellular (CSN) and Acellular Sciatic Nerve (ASN) Grafts

Nogo-A, MAG, and CSPG were detected by western blot in ON lysates and lysates from CSN and ASN grafts whilst Nogo-C was restricted to CSN and ASN grafts (Figure 5A,B). In addition, significantly lower levels of both Nogo-A and CSPG (*p* < 0.01) were observed in ASN grafts compared to CSN grafts (Figure 5A,B) suggesting that the freeze–thawing process may have removed some of these molecules. The levels of MAG and Nogo-C were unaffected by freeze–thawing.

These results suggest that significant levels of axon growth inhibitory molecules are present in both CSN and ASN grafts, which regenerating ON axons have to overcome.

### 3.5. Retinal Ganglion Cell (RGC) Axon Regeneration 21 Days After Grafting

RhB^+^ RGC axons were detected in the proximal optic nerve stump of all groups. Correlating with RGC survival, axon frequency in the proximal optic nerve segment was highest in the ivit_CSN_/ONT, ivit_S_/ONT_CSN_, ivit_CSN_/ONT_CSN_, and ivit_CSN_/ONT_ASN_ groups and least in the CON/ivit_S_/ONT, ivit_ASN_/ONT, and ivit_S_/ONT_ASN_ groups (not shown). (i), Growth into ONT: No RGC axons entered the distal optic nerve stump in CON/ivit_S_/ONT, ivit_ASN_/ONT, and ivit_CSN_/ONT groups (not shown) (ii), Growth into ONT_ASN_ grafts: No RGC axons grew into ONT_ASN_ nor after ivit_ASN_ (not shown). In the ivit_CSN_/ONT_ASN_ group, some RGC axons crossed the site of anastomosis site into the ONT_ASN_ graft, but very few penetrated farther than 400 μm (Figure 6A,C) and no optic nerve astrocyte processes entered the ONT_ASN_ grafts (Figure 6A inset). (iii), Growth into ONT_CSN_ grafts: No axons grew into ivit_S_/ONT_CSN_ grafts (not shown); however, in the ivit_CSN_/ONT_CSN_ group, there was almost a 5-fold increase in the number of axons extending 400 μm (*p* < 0.001) and significantly more axons regenerated for distances of 800 and 1200 μm within the ONT_CSN_ graft compared to ivit_CSN_/ONT_ASN_ groups (*p* < 0.001 and *p* < 0.001, respectively; Figure 6B,C). Astrocyte processes invaded the ONT_CSN_ for up to 500 µm (Figure 6B inset). Details of the ONT_CSN_ anastomosis site (Figure 6D,E) show the demarcation between the Schwann cell laminin^+^ basal lamina tubes of the CSN graft and optic nerve (Figure 6D). RhB^+^ axons (open arrowhead) associated with GFAP^+^ astrocyte processes (arrowhead), which had invaded the proximal region of the ONT_CSN_ and regenerated throughout the graft within Schwann cell basal lamina tubes (Figure 6E; arrows).

These results show that robust RGC axon regeneration is only promoted into ONT_CSN_ and not into ONT_ASN_ grafts by ivit_CSN_ implants, suggesting that: (1), Schwann cell and macrophage-derived trophic factors in the retina drive the regeneration of RGC axons into ONT_CSN_ and ONT_ASN_ grafts; (2) the growth of RGC axons within Schwann cell basal lamina tubes is not sustained in the absence of Schwann cells; (3), the additive axogenic effect of ivit_CSN_ and ONT_CSN_ seen in the ivit_CSN_/ONT_CSN_ compared to ivit_CSN_/ONT_ASN_ group is attributable to Schwann cell-derived factors originating from both sources.

### 3.6. Correlation of Retinal Ganglion Cell (RGC) Axon Regeneration with RIP of p75^NTR^ and Suppression of RhoGTP

(1) Regulated intramembranous proteolysis (RIP) of p75^NTR^: (i) in the retina: Levels of full-length p75^NTR^ (75kDa) and the extracellular domains (ECD), p75_ECD_ (≈55kDa), were up-regulated in ivit_S_/ONT, and ivit_ASN_/ONT_CSN_ compared to CON/intact lysates, but levels of the intracellular domain (ICD), p57_ICD_ (≈25kDa), were unchanged (Figure 7A,B) and, in the ivit_CSN_/ONT_CSN_ group, full-length p75^NTR^ was unchanged but both p75_ECD_ and p75_ICD_ were significantly elevated (*p* < 0.01 and *p* < 0.0001, respectively) compared to levels in the ivit_ASN_/ONT_CSN_ group (Figure 7A,B). (ii) In the optic nerve: Levels of full length and fragmented p75^NTR^ were similar in the ON of all groups, except in the ivit_CSN_/ONT_CSN_ group in which p75_ICD_ had increased significantly (*p* < 0.0001; Figure 7C,D).

(2) Suppression of Rho activation: (i) in the retina: There was significant activation of RhoGTP in the ivit_S_/ONT and ivit_ASN_/ONT_CSN_ groups compared to CON/intact groups, with higher levels present in the former than the latter, but activation was significantly reduced (*p* < 0.0001) in the ivit_CSN_/ONT_CSN_ group (Figure 7E). (ii) In the optic nerve: Levels of RhoGTP were elevated in the ON only in the CON/ivit_S_/ONT group and significantly reduced (*p* < 0.01) in the ivit_CSN_/ONT_CSN_ group (Figure 7E).

Thus, in the ivit_CSN_/ONT_CSN_ paradigm, p75_ECD_ shedding by RIP of p75^NTR^ with associated suppression of RhoGTP activity is likely to disable LIMK/cofilin-mediated axon growth cone collapse and correlate with enhanced RGC axon regeneration.

## 4. Discussion

We show here that Schwann cell-derived factor (SCDF)-rich CSN grafts transplanted into the vitreous enhance the survival of RGC after ONT, promoting the survival of 25% of RGC. The addition of a CSN graft, anastomosed to the cut end of the optic nerve, as well as an intravitreal CSN graft, was additive and promoted 58% RGC survival. In addition, intravitreal CSN grafts promoted the greatest activation of Müller cells/astrocytes and promoted ingress of macrophages into the retina. Intravitreal CSN combined with a CSN anastomosed to the cut end of the optic nerve promoted RGC axon regeneration into grafts, with axons in a CSN graft growing for longer distances than those in ASN grafts. The regenerative response to intravitreal CSN grafting and *ivit*_CSN_+ONT_CSN_ correlated with RIP of p75^NTR^ and suppressed RhoA activation, indicative of SCDF-dependent modulation of axon growth-inhibitory signalling.

### 4.1. The Effect of Schwann-cell-Derived Factor (SCDF) on the Survival of Axotomised RGC In Vivo

In the initial stages after optic nerve injury, NTF is supplied by reactive optic nerve astrocytes and macrophages/activated microglia [53,54,55], but the supply becomes exhausted, ultimately leading to RGC death. Consistent with this, RGC survival was reduced to <10% at 21 d after ONT, while ASN implanted intravitreally or anastomosed to the proximal optic nerve stump had no beneficial effects on axotomy-induced RGC death. In contrast, the grafting of an SCDF-rich CSN to the vitreous or vitreous plus the proximal ON stump enhanced RGC survival after ONT, supporting previous observations [31,33]. Intravitreal transplants of purified Schwann cells enhanced the survival of axotomised RGC in rats [56], confirming a role for SCDF in rat RGC survival. Schwann cells are a source of numerous NTF, including glial-derived neurotrophic factor (GDNF), nerve growth factor (NGF), brain-derived neurotrophic factor (BDNF), neurotrophin (NT)-4/5, and ciliary neurotrophic factor (CNTF) that can all promote RGC survival in vitro and in vivo [15,16,17,18]. RGC readily express tropomyosin receptor kinase (Trk)A, TrkB, and TrkC receptors and these are transiently up-regulated after optic nerve injury, increasing RGC responsiveness to NTF, thus promoting RGC survival immediately after injury [57]. An *ivit*_CSN_, however, enhanced the long-term expression of Trk receptors after axotomy [57] and hence, Schwann cells are likely to be responsible for the survival effects observed after CSN grafting in this study by: (1) SCDF signalling to rescue RGC from death; (2) enhancing Trk receptor expression, thereby enhancing NTF-mediated survival signals.

Our results also show that, although RGC survival is influenced by SCDF at the somata and axonal growth cones, the presence of Schwann cells in the vitreous promotes greater survival than their presence at the proximal stump of a transected optic nerve. This could imply a differential ability of SCDF to promote RGC survival when present at the somata or the injury site. It is possible that SCDF acting on RGC via growth cones require retrograde signalling to the somata, and are consequently less efficient at activating survival signals than SCDF acting on the somata directly.

### 4.2. The Effect of Sciatic Nerve (SN) Grafting on Müller Cells and Macrophages in the Retina

Schwann cells may indirectly influence the survival and regenerative responses of RGC axons, and it has previously been suggested that the response of RGC axons to intravitreal CSN grafting may be dependent on macrophage-derived NTF and/or stimulation of retinal glia to produce glial-derived NTF, which may, in turn, act upon RGC [49,58,59,60,61,62]. Retinal Müller cells express the NT receptors TrkB and TrkC, and produce NTF, including NGF, BDNF, NT-3, and CNTF [63,64]. Thus, NTF-activated Müller cells may produce NTF to support RGC survival and axon growth; however, the degree of Müller cell activation induced by *ivit*_CSN_ was significantly greater than that observed with *ivit*_ASN_ intravitreal SCDF activate Müller cell/astrocyte activation and that intravitreal SCDF-mediated RGC survival and axon regeneration is mediated partly through these activated retinal glia. Interestingly, the grafting of CSN to a transected optic nerve enhanced Müller cell activation above that observed after ASN grafting, suggesting that RGC effects mediated by SCDF at the injury site involve modulation by retinal glia.

After optic nerve injury, peripheral macrophages invade the injury site, and microglia become activated in both the retina and optic nerve, where they are transformed into phagocytic cells, termed microglia-derived macrophages. Activated microglia release potentially cytotoxic substances in vitro, including nitrogen monoxide, oxygen free-radicals, proteases, and excitatory amino acids [65], indicating that they accelerate neuronal degeneration; however, in addition to their phagocytic role in removing cell debris, both peripheral and microglia-derived macrophages secrete cytokines and NTF, including BDNF, GDNF [55], and oncomodulin [59]. The involvement of NTF secreted from intravitreal macrophages has been implicated in promoting RGC axon regeneration; retinal macrophage ingress has been proposed to play a role in the regenerative response to lens injury [58,59,60], and macrophage activation by intravitreal delivery of Zymosan, a yeast cell wall extract, supports axon regeneration [58,59,60].

The presence of macrophages in the vitreous and retina after *ivit*_CSN_ and not *ivit*_ASN_ grafting may suggest that these cells do play a role in the regenerative response observed in our present study. Macrophage-derived factors such as oncomodulin after inflammatory stimulation or lens injury have been shown to play a role in RGC survival and axon regeneration after ONT [49,50,51,52]; however, the role of oncomodulin in promoting RGC survival and axon regeneration is controversial and at best, limited [47,66]. In agreement with these latter studies, we detected no changes in oncomodulin mRNA over the first 7 days, in *ivit*_CSN_ paradigms where detected ED1^+^ macrophages were detected. This suggests that oncomodulin does not affect *ivit*_CSN_-mediated RGC survival and axon regeneration. In other studies, astrocyte-derived CNTF was shown to promote robust RGC axon regeneration after inflammatory stimulation and did not depend on oncomodulin or the presence of large numbers of activated macrophages in the retina [47,62]. In agreement with these studies, we detected significant levels of CNTF in the retina in the *ivit*_CSN_ paradigms, which may contribute to the net RGC survival and axon regenerative effects of CSN grafts.

### 4.3. The Effect of Schwann-Cell-Derived Factors (SCDFs) on Retinal Ganglion Cell (RGC) Axon Regeneration In Vivo

Despite the presence of intact laminin-rich basal lamina tubes in both CSN and ASN, RGC axons regenerated into SN grafts anastomosed onto the transected optic nerve, only in the presence of Schwann cells. These results support previous findings that the basal lamina alone is not capable of promoting RGC axon regeneration and that Schwann cells provide essential trophic support for axon growth [32,33,67]. Furthermore, RGC axons regenerated into an ASN graft after intravitreal CSN implantation, thus it can be concluded that SCDF, when delivered intravitreally, supports both axon growth within the inhibitory environment of the optic nerve and within the passive environment of the freeze–thawed ASN.

Axotomized RGC axons readily grow into artificial Schwann cell grafts anastomosed to the proximal transected optic nerve stump, further supporting a role for Schwann cells in promoting RGC axon regeneration [68]. Several NTF, particularly CNTF, produced by Schwann cells promote the growth of RGC axons. CNTF has been identified as a major RGC axogenic factor derived from SN [69] and CNTF promotes RGC neurite/axon growth in vitro [69,70,71] and in vivo [70,72,73]. In addition to diffusible NTF, cellular adhesion molecules expressed by Schwann cells may support axon growth into a CSN graft. RGC axons regenerating into a CSN graft are in direct contact with Schwann cells [67], probably through CAM interactions, and particularly the immunoglobulins L1 and neuronal cellular adhesion molecule have been associated with RGC regeneration into SN grafts [39]. Additionally, although the present study showed that components of the basal lamina alone do not promote RGC axon regeneration, ECM substrates such as laminin, collagen, and fibronectin support RGC axon growth during development [74], and may play a role in the guidance of regenerating axons.

There is an accumulation of evidence to indicate that the regenerative response of RGC to NTF is not simply a consequence of enhanced survival. In mice, overexpressing the anti-apoptotic protein Bcl-2, prevents axotomy-induced RGC death but RGC does not regenerate their axons [75]. Moreover, differential survival and regenerative responses are confirmed by the observation that intravitreal delivery of BDNF promotes the survival of RGC, but not the regeneration of their axons [76]. Likewise, we have shown previously that intravitreal inflammation elicited by intravitreal zymosan injection promotes RGC survival, as does intra-optic nerve zymosan injection but only intravitreal zymosan injection supports RGC axon regeneration [42]. These results suggest that although the survival cues derived from the retina and optic nerve may be the same, the cues for RGC axon regeneration are different may require an intravitreal glial component [42].

### 4.4. Schwann-Cell-Derived Factor (SCDF)-Induced Modulation of Growth-Inhibitory Signaling Molecules in the Retina and ON

The SN grafting experiments in this manuscript demonstrate that axons can regenerate through the inhibitory environment of a CSN graft when SCDFs are supplied to the RGC somata. One explanation may lie in the proposal that SCDFs enhance the expression and secretion of MMPs and modulate the production of tissue inhibitors of metalloproteases (TIMPs) in reactive astrocytes of the optic nerve [40]. MMPs are a family of enzymes that degrade proteinaceous ECM molecules, including fibronectin, laminin, collagen, and CSPG [77], thus causing dissolution of scar tissue and facilitating the passage of regenerating axons. In support of this proposal, RGC axon regeneration caused by *ivit*_CSN_ correlated with the absence of an inhibitory accessory glial limitans.

A second proposal is that SCDFs render growth cones insensitive to the growth inhibitory environment by down-regulating components of the axon growth-inhibitory signalling cascade [20,48]. For example, we have previously shown that the promotion of RGC axon regeneration by intravitreal SCDF after optic nerve crush injury causes RIP of p75^NTR^ and suppression of RhoA, thus disabling the RhoA-dependent axon growth-inhibitory cascade [20]. RIP of p75^NTR^ was mediated by NTF-mediated up-regulation of tumour necrosis factor-α converting enzyme (TACE) and γ-secretase [19,20], two enzymes that mediate RIP of p75^NTR^ [78,79,80]. The full-length p75^NTR^ molecule is processed into the 55kDa extracellular fragment (ECD) of p75^NTR^ (p75_ECD_) by tumour necrosis factor-α converting enzyme (TACE) which then activates γ-secretase to cleave the 25kDa intracellular domain (p75_ICD_) [48,80]. In the current study, we also observed similar RIP of p75^NTR^ after intravitreal SCDF (i.e., *ivit*_CSN_), yielding significantly increased levels of p75_ECD_ and p75_ICD_ in both retinae and optic nerve, and correlating with suppressed RhoGTP levels. We suggest that RGC axon growth into a CSN graft, driven by SCDF in an *ivit*_CSN_ graft is also mediated through RIP of p75^NTR^ and thus blinding of axons to the inhibitory environment of the distal nerve [19,20].

## 5. Conclusions

We show that SCDFs from CSN grafts are required to support RGC survival after ONT and that RGC survival can further be enhanced using a CSN graft anastomosed onto the cut end of the transected optic nerve. In addition, limited numbers of RGC axons regenerate into an ASN graft after ONT but RGC axon regeneration is significantly enhanced when an *ivit*_CSN_ is combined with an ONT_CSN_ graft. Moreover, RGC axon regeneration promoted by an *ivit*_CSN_/ONT_CSN_ causes RIP of p75^NTR^, suppressed RhoA activation, and thus disablement of the axon growth-inhibitory cascade, blinding axons to potential myelin-derived axon growth-inhibitory ligands.

## Figures and Tables

**Figure 1 cells-09-01335-f001:**
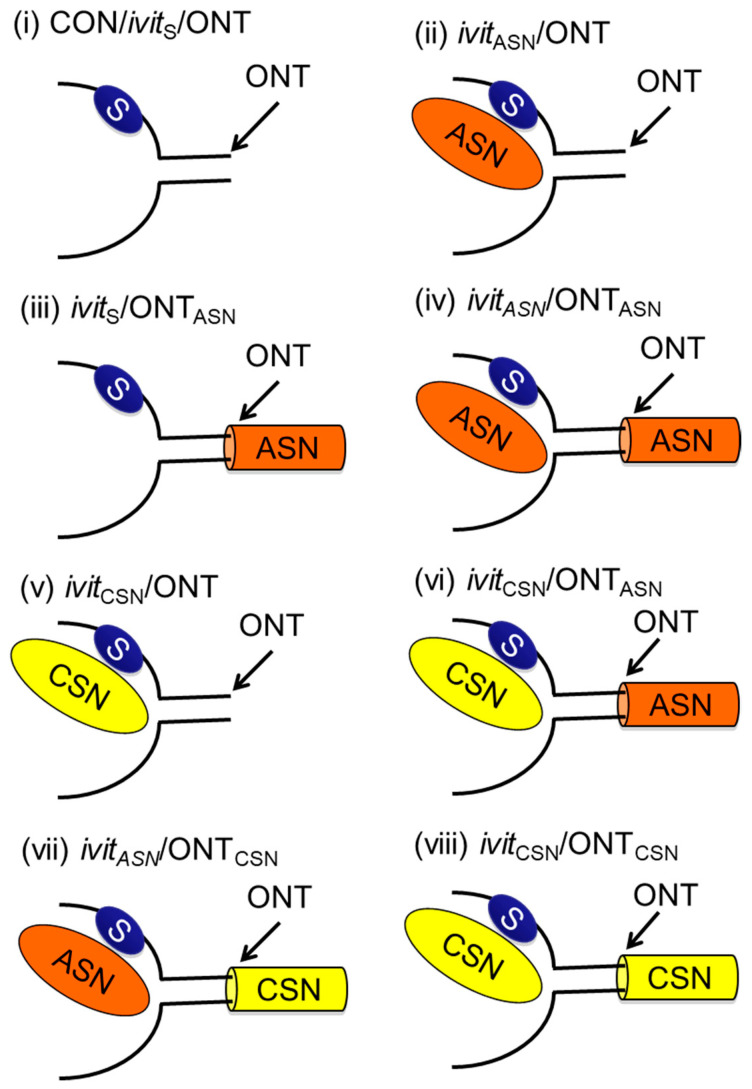
Control and experimental groupings (see text). (**i**) Control (CON)/intravitreal (*ivit*) Sterispon (_S_)/optic nerve transection (ONT), (**ii**) *ivit* acellular sciatic nerve grafts (_ASN_)/ONT; (**iii**) *ivit*_S_/ONT_ASN_; (**iv**) *ivit*_ASN_/ONT_ASN_; (**v**) *ivit* cellular sciatic nerve grafts (_CSN_)/ONT; (**vi**) *ivit*_CSN_/ONT_ASN_; (**vii**) *ivit*_ASN_/ONT_CSN_; (**viii**) *ivit*_CSN_ONT_CSN_.

**Figure 2 cells-09-01335-f002:**
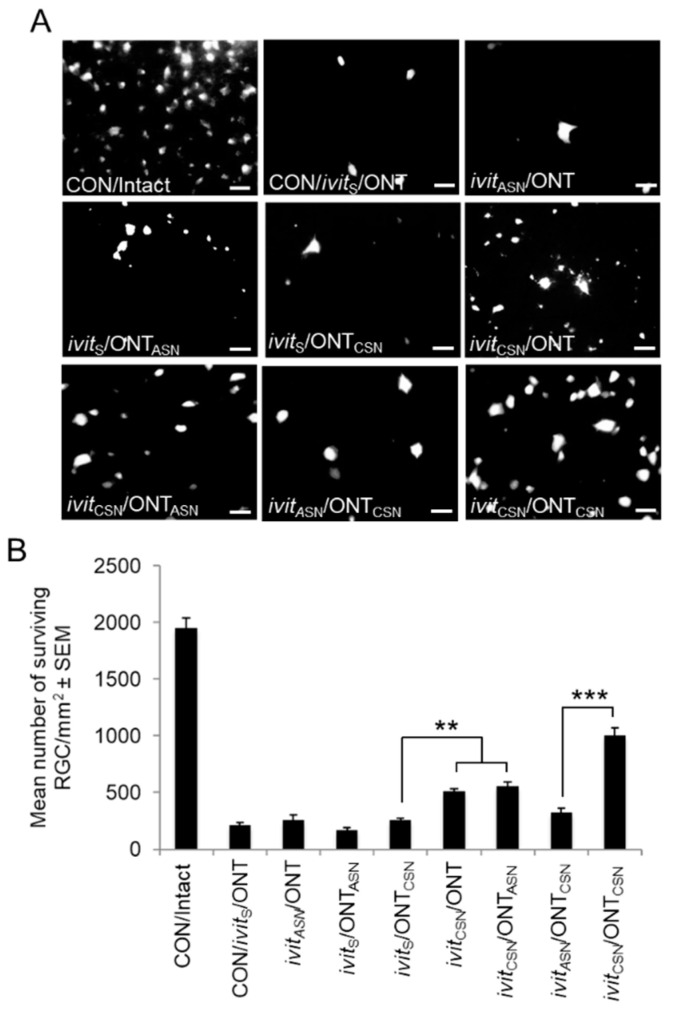
Fluorogold (FG)^+^ retinal ganglion cells (RGC) at 21 days after grafting in retinal wholemounts from: (**A**) CON/intact, CON/*ivit*_S_/ONT, *ivit*_ASN_/ONT, *ivit*_S_/ONT_ASN_, *ivit*_S_/ONT_CSN_, *ivit*_CSN_/ONT, *ivit*_CSN_/ONT_ASN_, *ivit*_ASN_/ONT_CSN_, and *ivit*_CSN_/ONT_CSN_ groups. Note that the *ivit*_CSN_/ONT_CSN_ group show the greatest RGC neuroprotection (scale bars = 20 μm). (**B**) Quantification of numbers of FG^+^ RGC in retinal wholemounts confirms that RGC survival is increased in all *ivit*_CSN_ paradigms with the greatest survival observed in the *ivit*_CSN_/ONT_CSN_ group. *n* = 16 retinae/group; ** *p* < 0.001; *** *p* < 0.0001, ANOVA.

**Figure 3 cells-09-01335-f003:**
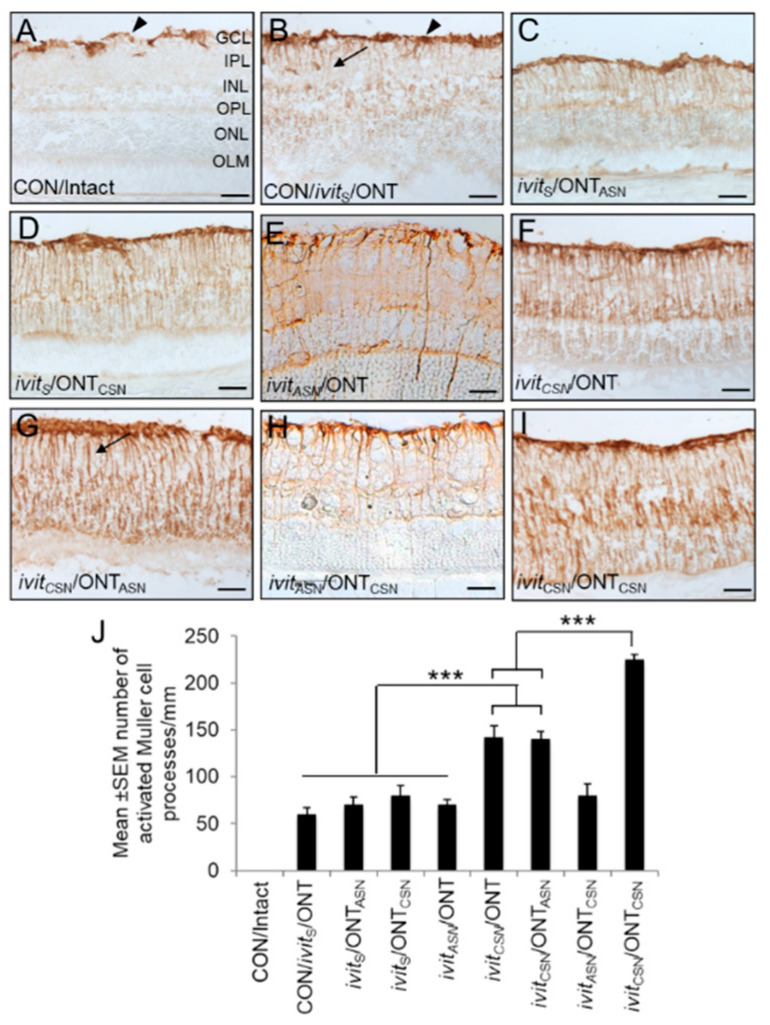
Müller cell/astrocyte activation in the retina at 21 days after grafting. Representative images of glial fibrillary acidic protein (GFAP)^+^ astrocytes (arrowheads) and activated Müller cell processes (arrows) in: (**A**) CON/intact, (**B**) CON/*ivit*_S_/ONT, (**C**) *ivit*_ASN_/ONT, (**D**) *ivit*_S_/ONT_ASN_, (**E**) *ivit*_S_/ONT_CSN_, (**F**) *ivit*_CSN_/ONT, (**G**) *ivit*_CSN_/ONT_ASN_, (**H**) *ivit*_ASN_/ONT_CSN_, and (**I**) *ivit*_CSN_/ONT_CSN_ groups. Note the increased glial activation after *ivit*_CSN_ grafting (scale bars = 50 μm). (**J**) Quantification of the number of Müller cell processes confirms that *ivit*_CSN_ promotes glial activation in the retina. *n* = 16 retinae/group; *** *p* < 0.0001, ANOVA.

**Figure 4 cells-09-01335-f004:**
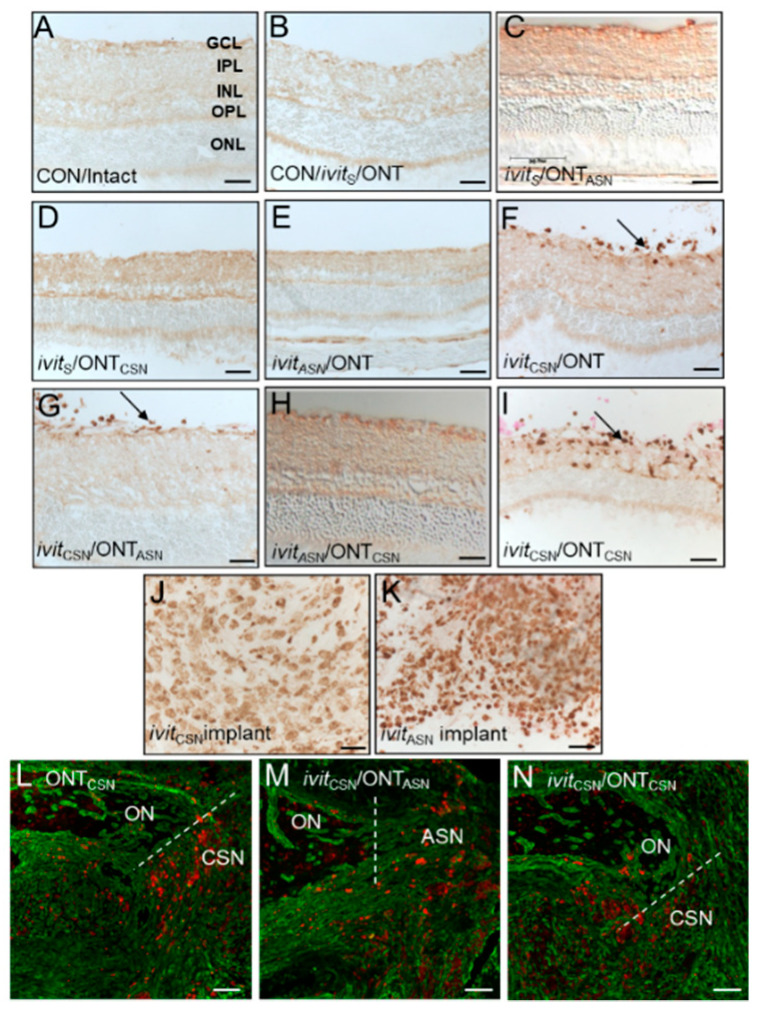
Macrophages accumulate in the retina in eyes containing *ivit*_CSN_ implants at 21 days after grafting. ED1-HRP^+^ macrophages in the vitreous body and retina in: (**A**) CON/intact, (**B**) CON/*ivit*_S_/ONT, (**C**) *ivit*_ASN_/ONT, (**D**) *ivit*_S_/ONT_ASN_, (**E**) *ivit*_S_/ONT_CSN_, (**F**) *ivit*_CSN_/ONT, (**G**) *ivit*_CSN_/ONT_ASN_, (**H**) *ivit*_ASN_/ONT_CSN_, and (**I**) *ivit*_CSN_/ONT_CSN_ groups. ED1^+^ macrophages were present in both (**J**) *ivit*_CSN_, and (**K**) *ivit*_ASN_ implants; ED1^+^ macrophages (red) at the anastomosis site of (**L**) ONT_CSN_; (**M**) *ivit*_CSN_/ONT_ASN_; (**N**) *ivit*_CSN_/ONT_CSN_ groups (scale bars in **A**–**K** = 50 μm; in **L**–**N** = 100 μm). *n* = 16 retinae/group.

**Figure 5 cells-09-01335-f005:**
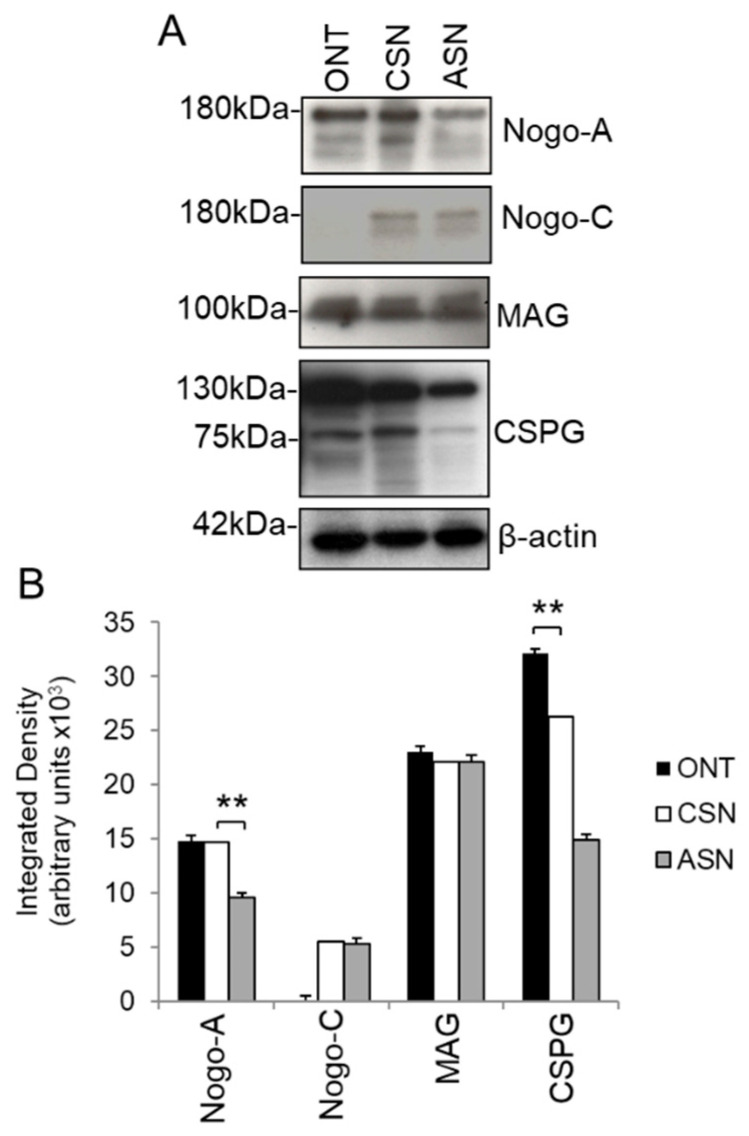
Growth inhibitory ligands in the optic nerve, cellular sciatic nerve (CSN), and acellular sciatic nerve (ASN) grafts at 21 days after grafting. (**A**) Nogo-A, MAG, and CSPG were all present in optic nerve, CSN, and ASN whilst Nogo-C was not present in optic nerve. (**B**) Densitometry to confirm the levels of Nogo-A, Nogo-C, MAG, and CSPG in optic nerve, ASN, and CSN lysates (*n* = 6/per group, 3 independent repeats). ** *p* < 0.001, ANOVA. β-actin was used as a loading control.

**Figure 6 cells-09-01335-f006:**
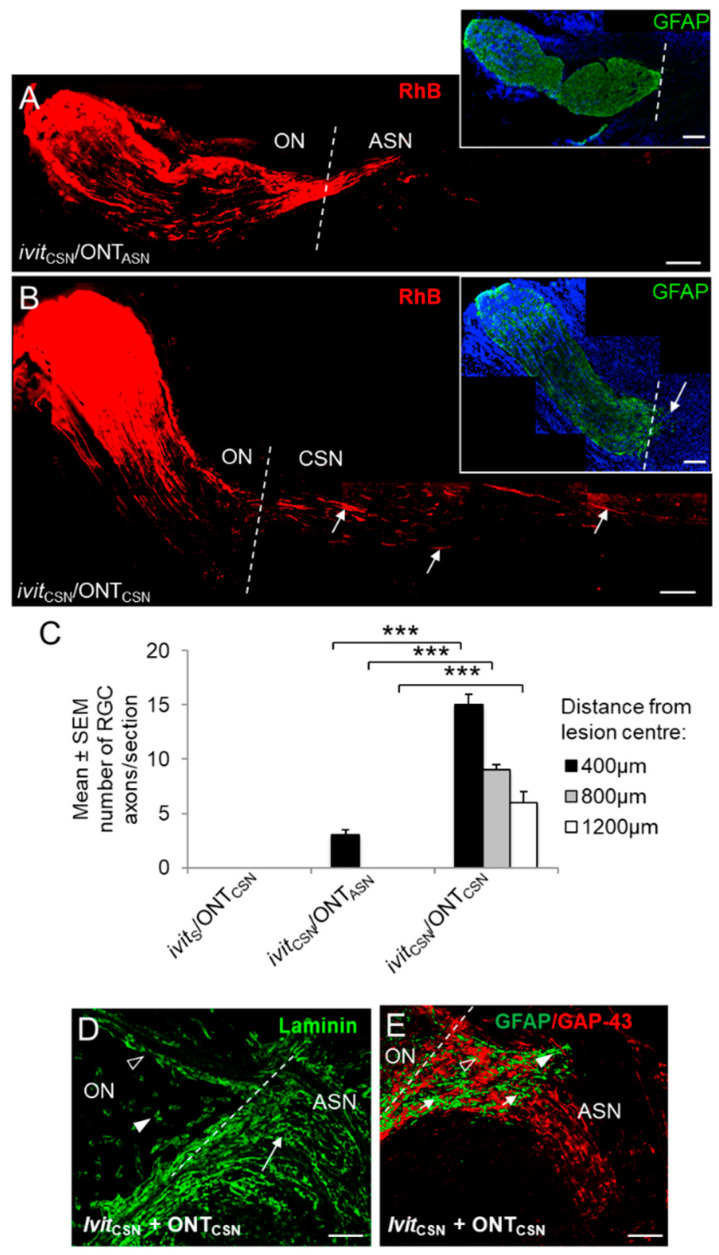
Regeneration of rhodamine B (RhB)^+^ RGC axons (red) into cellular sciatic nerve (CSN) or acellular sciatic nerve (ASN) grafts at 21 days after optic nerve transection (ONT). Representative images of longitudinal sections of optic nerve and graft in (**A**), *ivit*_CSN_/ONT_ASN_ and (**B**), *ivit*_CSN_/ONT_CSN_ groups—note the presence throughout the length of the ONT_CSN_ graft of RhB^+^ (red) regenerating RGC axons (open arrows) and their limited penetration into the ONT_ASN_ graft (**A**,**B** insets), optic nerve GFAP^+^ astrocyte processes (green) do not enter ONT_ASN_ grafts, but do enter the proximal region of the ONT_CSN_ graft (closed arrow) (scale bars in **A**, **B**, and **D** = 200 µm; broken line indicates anastomosis site). (**C**) Quantification of the number of axons extending 400, 800, and 1200 μm past the anastomosis site into the ONT_CSN_ graft (*** *p* < 0.0001). A few axons penetrate the junctional zone of ONT_ASN_ grafts compared to the numbers extending for long distances into ONT_CSN_ grafts. (**D**), anastomosis site in an *ivit*_CSN_/ONT_CSN_ rat showing the laminin^+^ basal lamina of the optic nerve vasculature (arrow heads) and Schwann cell basal lamina tubes aligned in parallel arrays in the ONT_CSN_ graft (arrow), with which (**E**) regenerating RGC axons are associated with the extent of GFAP^+^ astrocyte process (green) invasion and the growth of RhB^+^ RGC axons (red) (scale bars = 150 µm; the broken line indicates the anastomosis site).

**Figure 7 cells-09-01335-f007:**
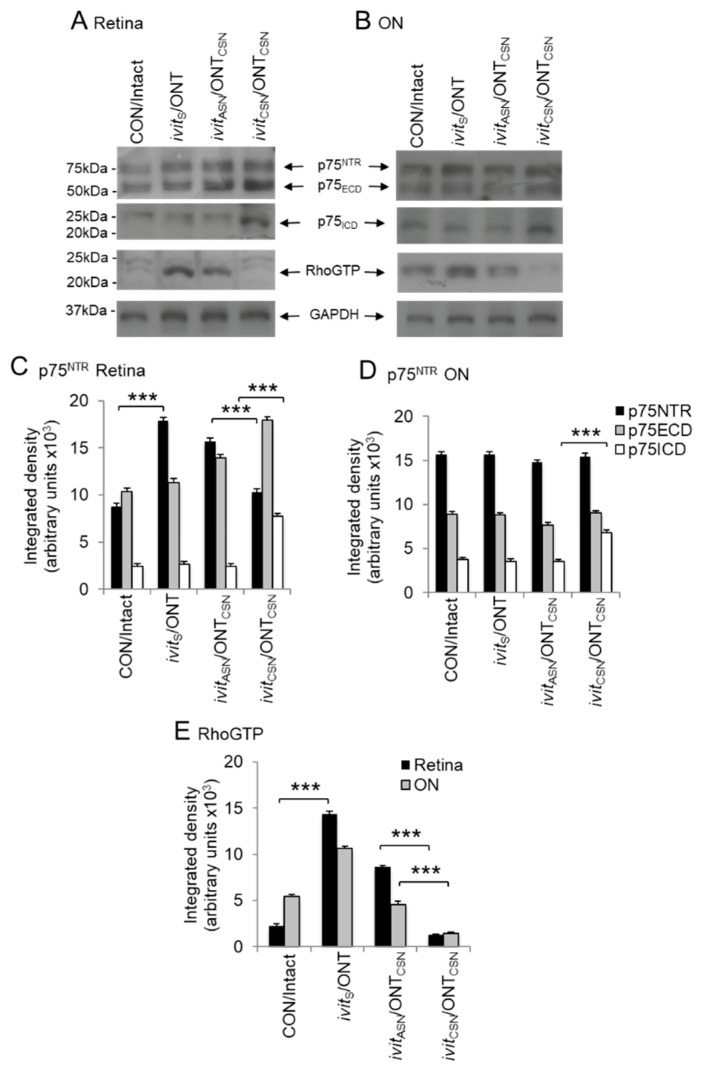
Axon regeneration promoted by *ivit*_CSN_ is correlated with reduced levels of RhoGTP and regulated intramembranous proteolysis (RIP) of p75^NTR^ in the optic nerve and retina. In both the retina (**A**,**C**) and optic nerve (**B**,**D**) p75 intracellular domain (p75_ICD_) accumulates only in *ivit*_CSN_/ONT_CSN_ group. RhoGTP is activated in the retina and optic nerve and remains high in *ivit*_ASN_/ONT_CSN_ group and significantly suppressed in *ivit*_CSN_/ONT_CSN_ group (**E**). β-actin was used as a loading control. *** *p* < 0.0001, *n* = 8 retinae/ON pooled/treatment, three independent repeats).

**Table 1 cells-09-01335-t001:** List of antibodies used in this study.

Antibody	Specificity	Dilution	Source
Primary Antibodies			
Mouse anti-cluster differentiation 68 (ED1)	Rat macrophages	1:200	Serotec, Kidlington, UK
Mouse anti-glial fibrillary acidic protein (GFAP)	Astrocyte	1:400	Sigma, Poole, U
Mouse anti-growth associated protein-43 (GAP-43)	Regenerating axons	1:500	Invitrogen, Paisley, UK
Mouse anti-myelin associated glycoprotein (MAG)	MAG	1:400	Abcam, Cambridge, UK
Mouse anti-chondroitin sulphate proteoglycans (CSPG)	CSPG	1:500	Abcam, Cambridge, UK
Rabbit anti-laminin	Basal lamina/glia limitans and blood vessels	1:500	Sigma, Poole, UK
Rabbit anti-p75 neurotrophic receptor	Low affinity neurotrophin receptor/Schwann cells	1:400	Sigma, Poole, UK
Rabbit anti-neurite outgrowth inhibitor A (Nogo-A)	Nogo-A	1:400	Abcam, Cambridge, UK
Sheep anti-Nogo-C	Nogo-C	1:500	R&D systems Europe, Oxford, UK
Secondary Antibodies			
Alexa 488 anti-mouse	Mouse IgG	1:400	Invitrogen, Paisley, UK
Texas Red anti-mouse	Mouse IgG	1:400	Invitrogen, Paisley, UK
Alexa 488 anti-rabbit	Rabbit IgG	1:400	Invitrogen, Paisley, UK
Texas Red anti-rabbit	Rabbit IgG	1:400	Invitrogen, Paisley, UK
HRP-labelled anti-rat	Rat IgG	1:400	Vector Laboratories, Peterborough, UK
HRP-labelled anti-rabbit	Rabbit IgG	1:400	Vector Laboratories, Peterborough, UK
HRP-labelled anti-sheep	Sheep IgG	1:400	Invitrogen, Paisley, UK

**Table 2 cells-09-01335-t002:** Retinal ganglion cell (RGC) survival after treatment with or without cellular (CSN) or acellular (ASN) intravitreal (*ivit*) sciatic nerve grafts after optic nerve transection (ONT). (** *p* < 0.01, *** *p* < 0.0001, ANOVA).

Condition	Group	RGC Count	RGC Death	RGC Survival
Controls	CON/Intact	1953 ± 84		
	CON/*ivit*_S_/ONT	211 ± 10	89%	11%
*Ivit* grafts	*ivit*_CSN_/ONT	510 ± 23	75%	25% **
	*ivit*_ASN_/ONT	171 ± 55	91%	9%
ON grafts	*ivit*_S_/ONT_CSN_	255 ± 21	87%	13%
	*ivit*_S_/ONT_ASN_	170 ± 22	91%	9%
*Ivit*+ON grafts	*Ivit*_ASN_/ONT_CSN_	325 ± 35	83%	17%
	*ivit*_CSN_/ONT_ASN_	552 ± 40	72%	28% **
	*ivit*_CSN_/ONT_CSN_	1000 ± 70	42%	58% ***

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
