# Peer review of "Retinal Ganglion Cell Survival and Axon Regeneration after Optic Nerve Transection is Driven by Cellular Intravitreal Sciatic Nerve Grafts"

_cells, 2020, doi:10.3390/cells9061335_

Round 1

Reviewer 1 Report

In this study, the authors test the effects of transplanting short portions of sciatic nerve in the eye or onto the optic nerve following an optic nerve crush. They use either fresh sciatic nerve, or ones that had been through several freeze-thaw cycles to kill Schwann cells. This is an attempt to test the influence of how neurotrophic factors released by Schwann cells affect survival of retinal ganglion cells (RGC) and re-growth of RGC axons. The major finding is that intravitreal transplantation of the fresh sciatic nerve along with transplantation onto the injured optic nerve enhances RGC survival and promotes RGC axon regeneration. The manuscript is quite thorough and detailed and the conclusions seem sound based on the presented data. My only critique is that this manuscript suffers a bit from extensive use of abbreviations that make reading and interpretation a challenge and that the manuscript could be strengthened by streamlining the prose. In some ways, however the abbreviations are necessitated by the complex array of experimental conditions and Figure 1 helps as a good reference to understand them. 

Author Response

"The manuscript is quite thorough and detailed and the conclusions seem sound based on the presented data. My only critique is that this manuscript suffers a bit from extensive use of abbreviations that make reading and interpretation a challenge and that the manuscript could be strengthened by streamlining the prose. In some ways, however the abbreviations are necessitated by the complex array of experimental conditions and Figure 1 helps as a good reference to understand them." 

Author response: We have removed some of the abbreviations to make it easier to follow. These include: ON (optic nerve), OM (oncomodulin), NO (nitrogen monoxide), CAMs (cellular adhesions molecules) and NCAM. Other abbreviations cannot be removed due to multiple use and explains he experimental groups which are complicated, as acknowledged by this reviewer.

We have checked the sentences in the text and unfortunately there is nothing that can be changed in terms of streamlining the prose. We refer you to reviewer 2 comments below who says “Overall, this is a well written article”. Hence, we feel further changes are not required.  

Reviewer 2 Report

In this manuscript Ahmed et. al., studied the neuroprotective and axon growth effect of Schwann cells (SCs) in retinal ganglion cell (RGC) neurons. The authors used sciatic nerve (SN) grafts transplanted into the vitreous body of the eye, or anastomosis to the distal stump of a transected optic nerve, or did both. To identify the roles of SCs in neuroprotection and axon growth, the authors used SN grafts in which, SCs had been killed (ASN) or remained intact (CSN). Their results showed that placement of CSN at intravitreal (ivit) site or anastomosed to optic nerve (ON) promote RGC survival. They further report that when CSN is simultaneously placed in both sites, they exert additive RGC neuroprotection. Moreover the authors further show this neuroprotective and axon growth effect is lost when they used ASN.  Finally the authors show data and conclude that, RGC axon regeneration promoted by ivitCSN/ONTCSN causes RIP of p75NTR, suppressed RhoA activation and thus disablement of the axon growth inhibitory cascade, blinding axons to potential myelin-derived axon growth inhibitory ligands.

Overall this is a well written article, and the conclusions are supported by well controlled experiments. 

Author Response

"Overall, this is a well written article, and the conclusions are supported by well controlled experiments." 

Author response: There is nothing to respond to from this reviewer. We are very grateful for the reviewer’s time and effort in reviewing our manuscript.